# A Humanized Yeast Model for Studying TRAPP Complex Mutations; Proof-of-Concept Using Variants from an Individual with a *TRAPPC1*-Associated Neurodevelopmental Syndrome

**DOI:** 10.3390/cells13171457

**Published:** 2024-08-30

**Authors:** Erta Zykaj, Chelsea Abboud, Paria Asadi, Simane Warsame, Hashem Almousa, Miroslav P. Milev, Brittany M. Greco, Marcos López-Sánchez, Drago Bratkovic, Aashiq H. Kachroo, Luis Alberto Pérez-Jurado, Michael Sacher

**Affiliations:** 1Department of Biology, Concordia University, Montreal, QC H4B1R6, Canada; ertazykaj1987@gmail.com (E.Z.); chelsea.abboud@lau.edu (C.A.); paria.asadi@mail.concordia.ca (P.A.); warsame.simane@gmail.com (S.W.); hashem.almousa@gmail.com (H.A.); miroslav.milev@mail.mcgill.ca (M.P.M.); brittany.greco@concordia.ca (B.M.G.); aashiq.kachroo@concordia.ca (A.H.K.); 2Department of Medicine and Life Sciences, Universitat Pompeu Fabra, 08003 Barcelona, Spain; lpzsnchzmarcos@gmail.com (M.L.-S.); luis.perez@upf.edu (L.A.P.-J.); 3Hospital del Mar, Hospital del Mar Research Institute (IMIM), 08003 Barcelona, Spain; 4Centro de Investigación Biomédica en Red de Enfermedades Raras (CIBERER), ISCIII, 28029 Madrid, Spain; 5Women’s and Children’s Hospital, Metabolic Clinic, North Adelaide, SA 5006, Australia; drago.bratkovic@sa.gov.au; 6Department of Anatomy and Cell Biology, McGill University, Montreal, QC H3A 0C7, Canada

**Keywords:** yeast, TRAPP, TRAPPC1, mutation, humanization, Golgi, autophagy

## Abstract

Variants in membrane trafficking proteins are known to cause rare disorders with severe symptoms. The highly conserved transport protein particle (TRAPP) complexes are key membrane trafficking regulators that are also involved in autophagy. Pathogenic genetic variants in specific TRAPP subunits are linked to neurological disorders, muscular dystrophies, and skeletal dysplasias. Characterizing these variants and their phenotypes is important for understanding the general and specialized roles of TRAPP subunits as well as for patient diagnosis. Patient-derived cells are not always available, which poses a limitation for the study of these diseases. Therefore, other systems, like the yeast *Saccharomyces cerevisiae*, can be used to dissect the mechanisms at the intracellular level underlying these disorders. The development of CRISPR/Cas9 technology in yeast has enabled a scar-less editing method that creates an efficient humanized yeast model. In this study, core yeast subunits were humanized by replacing them with their human orthologs, and TRAPPC1, TRAPPC2, TRAPPC2L, TRAPPC6A, and TRAPPC6B were found to successfully replace their yeast counterparts. This system was used for studying the first reported individual with an autosomal recessive disorder caused by biallelic *TRAPPC1* variants, a girl with a severe neurodevelopmental disorder and myopathy. We show that the maternal variant (TRAPPC1 p.(Val121Alafs*3)) is non-functional while the paternal variant (TRAPPC1 p.(His22_Lys24del)) is conditional-lethal and affects secretion and non-selective autophagy in yeast. This parallels defects seen in fibroblasts derived from this individual which also showed membrane trafficking defects and altered Golgi morphology, all of which were rescued in the human system by wild-type *TRAPPC1*. This study suggests that humanized yeast can be an efficient means to study TRAPP subunit variants in the absence of human cells and can assign significance to variants of unknown significance (VUS). This study lays the foundation for characterizing further TRAPP variants through this system, rapidly contributing to disease diagnosis.

## 1. Introduction

Rare diseases are considered those that affect 200,000 individuals or fewer in the United States [1] and affect 3.5–5.9% of the world population [2], with an estimation of around 30 million affected people in the U.S. and E.U. [3,4]. Despite their low prevalence individually, more than 10,000 rare diseases are known worldwide [5], and with the rapid improvement of genetic technologies, new diseases are continuously being identified [2]. Rare variant analysis through exome sequencing in families with individuals who suffer from undiagnosed diseases is a potent method to discover ‘orphan’ gene versions [6]. While it is difficult to define the precise cause of a rare disorder, most of them are generated by single-gene mutations [4]. The burden that rare conditions pose for the patients, caregivers, and society is tremendous [7,8]. Because of their chronic and degenerative nature, these diseases extensively affect quality of life. In addition, rare disorders take time to diagnose, particularly as new genes are identified and novel variants in the new and known candidate genes are classified as variants of unknown significance (VUS). Such variants pose scientific and ethical dilemmas for clinicians, researchers, and families [9].

The compartmentalization of eukaryotic cells confines metabolic activities into individual organelles. Macromolecules of the endomembrane system are transported to their proper destination in a process referred to as membrane trafficking [10]. An increasing number of disorders linked to defects in membrane traffic proteins have been reported in recent years and the number of genes known to cause these predominantly monogenic disorders has grown from 49 genes in 2011 to over 300 within a decade [11]. Diseases caused by defects in these proteins frequently involve multiple systems but can also be manifested as more tissue or organ-specific [12]. Approximately 70% of rare monogenic disorders are manifested with neurodevelopmental symptoms [13,14]. By studying these rare Mendelian disorders, we can gain important insights into the function of membrane traffic genes and their complex molecular interactions.

TRAPP complexes are multi-subunit tethering factors initially identified in the yeast *Saccharomyces cerevisiae* that are highly conserved throughout eukaryotes. Two forms of the complex called TRAPPP II and TRAPP III have been reported in both yeast and human cells. Both complexes in yeast are composed of a common ‘core’ of subunits that include Bet5p, Trs20p, Bet3p, Trs23p, Trs31p, and Trs33p [15]. With the exception of Trs33p, all of the other core subunits are essential for cell viability [16]. In addition to this core, TRAPP II has four additional subunits (Tca17p, Trs65p, Trs120p, and Trs130p) [17,18,19,20], while TRAPP III has one additional subunit (Trs85p) [21]. Although the exact function of these extra subunits is still ambiguous, they are thought to stabilize the complexes by giving them substrate and compartmental specificity [22]. Although their role as bona fide tethers remains to be proven, it is thought that they carry out their function indirectly through their activity as Rab guanine nucleotide exchange factors (GEF) [23]. TRAPP II is known to function in the late Golgi, regulating processes like endocytosis and post-Golgi trafficking [19,24], while TRAPP III is implicated in ER-to-Golgi trafficking and autophagy [25,26,27]. The latter complex mediates its functions in ER-to-Golgi traffic and autophagy by activating the small GTPase RAB1/Ypt1p. This GTPase recruits distinct effectors for each process, including p115 for membrane trafficking and ATG1 for autophagy [28,29]. Acting downstream of these effectors are SNARE proteins in membrane trafficking and ATG8/LC3 in autophagy. The TRAPP II complex also activates a GTPase called RAB11/Ypt31/32p with a distinct set of effectors. Thus, TRAPP complexes play early roles in these processes, at the level of GTPase activation.

Like their yeast counterpart, human TRAPP complexes are also composed of a common core of proteins that include TRAPPC1 (Bet5p ortholog), TRAPPC2 (Trs20p ortholog), TRAPPC2L (Tca17p ortholog), TRAPPC3 (Bet3p ortholog), TRAPPC4 (Trs23p ortholog), TRAPPC5 (Trs31p ortholog), and TRAPPC6A/C6B (Trs33p ortholog). Other than the core, TRAPP II includes TRAPPC9 (Trs120p ortholog) and TRAPPC10 (Trs130p ortholog), while TRAPP III includes TRAPPC8 (Trs85p ortholog), TRAPPC11, TRAPPC12, and TRAPPC13 (Table 1) [30], where TRAPPC11 and TRAPPC12 are metazoan specific subunits [20].

The detrimental effects of mutations in genes that encode TRAPP subunits are proof of the importance of TRAPP complexes in humans [31]. In recent years, numerous mutations have been reported in TRAPP genes that are collectively termed TRAPPopathies, and the number of diseases associated with these complexes is expanding [15]. Thus far, mutations in several core TRAPP proteins, including TRAPPC2 [32], TRAPPC4 [33,34], and TRAPPC6A/B [35,36,37,38], have been reported and presumably affect the functions of both TRAPP II and III complexes. Mutations in proteins specific to TRAPP II and III have also been reported [39,40,41,42,43,44,45,46]. Disorders caused by variants in TRAPP proteins fall mainly into three categories: neurodevelopmental disorders, muscular dystrophies, and skeletal dysplasias [15,32,42].

One of the simplest sources for studying cellular phenotypes related to rare diseases is human dermal fibroblasts (HDF) [47]. These are easily obtained and maintained and are genetically stable [48,49]. HDFs have been used in many studies as cellular models for characterizing mutations that are linked to TRAPPopathies [39,43,50]. Although patient-derived cells make an ideal model system for studying the molecular mechanism of many diseases, the procedure for obtaining these cells is invasive. Therefore, the number of patients who agree to undergo a skin biopsy is relatively small [51]. This poses a limitation when it comes to characterizing disease-causing mutations in humans and analyzing VUS. Thus, other approaches that do not rely on patient cells offer new possibilities for studying these disorders.

*Saccharomyces cerevisiae* has long been used as a model system for studying human biology due to its evolutionarily conserved genes and pathways that are very similar to those in humans [52,53]. Roughly 30% of disease-associated genes in humans have orthologs in yeast and can functionally replace their yeast counterparts [54]. An efficient way to use yeast as an experimental tool to study human diseases is by humanizing yeast cells. Laurent and colleagues considered five degrees of yeast humanization: (degree 0) non-humanized yeasts as systems for analyzing human-specific processes, (degree 1) heterologous expression of human genes in yeast, (degree 2) humanization of specific sites in yeast genes, (degree 3) humanization of complete yeast genes, (degree 4) and humanization of entire pathways in yeast [55].

One way of studying human mutations in yeast is the replacement of the entire yeast gene with the open reading frame (ORF) of its human ortholog. In the last 30 years, there have been many attempts to humanize yeast genes [53,55,56,57], mainly due to the simplicity and effectiveness of this process [52]. The development of CRISPR/Cas9 technology in yeast has proved to be an extremely useful tool [58]. Unlike conventional methods, it has enabled the efficient replacement of yeast genes with their human orthologs in a scar-less way and with no need for a selectable marker [59]. This approach consists of generating single plasmids that express both the Cas9 nuclease and the guide RNA, which will target any desired locus in the yeast genome. These plasmids, when transformed together with a repair template (i.e., human ORF), enable the efficient replacement of yeast genes with their human orthologs [59,60].

The main objective of this study is to create a humanized yeast strain that can be used as a platform to study TRAPP core subunit variants, particularly VUS, especially when there are restrictions in obtaining cells from an affected individual. Using the CRISPR/Cas9 editing tool, we show that several yeast TRAPP core genes can be singly humanized while others cannot. We then use this system to characterize the first known *TRAPPC1* gene variants. Since the affected individual has compound heterozygous VUS, our system was able to reveal that both variants compromise function, moving these variants into the realm of likely disease-causing. Additionally, defects at the yeast cellular level with the humanized yeast are similar to those seen in fibroblasts derived from the affected individual, confirming this yeast system as a useful model system to study TRAPP gene variants when human cells are not available.

## 2. Materials and Methods

A complete list of plasmids and yeast strains used or created in this study are presented in Table 2 and Table 3, respectively.

### 2.1. Preparation of sgRNAs That Target the Yeast TRAPP Genes

All the single guide RNA oligonucleotides (sgRNA) targeting the yeast TRAPP genes were designed using the Geneious software (2024.0) and the Doench algorithm to score sgRNA for high on-target and low off-target activity [61,62]. The sequences used are shown in Table 4.

Forward and reverse oligonucleotides with the sgRNA sequence and Golden Gate compatible overlaps were mixed (10 µM each) in a total volume of 20 µL and annealed with each other using a thermocycler programmed to start at 95 °C and reduce the temperature in 5 °C increments every 15 min. An amount of 2 µL of annealed oligonucleotides (1:500 dilution) was used in the Golden Gate reaction for the assembly of the sgRNA cassette into the CRISPR/Cas9 plasmid (pCEN6-Cas9-GFP-KanMX).

### 2.2. Preparation of Repair Template DNA

The repair templates (human ORFs) were amplified using oligonucleotides designed for each of the human TRAPP genes that contained flanking sequences from the yeast genome. The sequences are shown in Table 5.

To obtain 5 µg or more of the repair template DNA needed, PCR reactions of 150 µL were set using a high-fidelity polymerase (Phusion, New England Biolabs, Ipswich, MA, USA). Each template PCR was verified with agarose gel electrophoresis and then purified to use for yeast transformation.

### 2.3. Yeast Transformations

Yeast strains were transformed using the Frozen-EZ Yeast Transformation II Kit (Zymo Research, Irvine, CA, USA) as per the manufacturer’s protocol. An amount of 250 µL was plated on selective media (YPD + G418).

### 2.4. Primer Design for Confirmation via Colony PCR

PCR primers to confirm gene replacement were designed for each of the human ORFs replacing their yeast counterparts such that the forward primers annealed to the yeast 5’ untranslated region (UTR) and the reverse primers annealed only to the new human ORF. The oligonucleotides are listed in Table 6.

### 2.5. Colony Screening via PCR

Colonies from YPD + G418 plates were picked using a pipette tip, suspended in 50 μL of water, and microwaved at the highest power for 2 min to lyse the cells for a genomic DNA (gDNA) template. Using Phire Plant Direct PCR Master Mix Kit (Thermo Fisher Scientific, Waltham, MA, USA) as the polymerase, a 10 μL PCR reaction was set up, with confirmation primers and 2 μL of the gDNA template. The colony PCR reactions were fractionated by agarose gel electrophoresis to assess successful replacement of the yeast gene with its human ortholog.

### 2.6. Yeast Liquid Growth Assays

A BioTek Synergy HT incubating spectrophotometer (BioTek Instruments, Winooski, VT, USA) was used to perform the liquid growth assays. Yeast cultures of 150 µL were seeded in a 96-well plate with an initial cell density of 2.5–5 × 10^5^ cells/mL by normalizing them to approximately 0.01–0.02 OD600 units, across three replicates. The cultures were grown and OD600 was measured in the microplate reader, with continuous double orbital shaking. Readings were taken every 10–20 min over a 30 h time period.

### 2.7. Autophagy Assays in Yeast

Processing of pre-Ape1p was assessed by monitoring different forms of the protein using Western blot analysis [63]. Yeast cultures of 10 mL were grown overnight to an early-to-mid-log phase at OD600 = 0.5–0.8 in YPD at 30 °C. Five OD units of cells were pelleted at 2500 rpm in a tabletop centrifuge for 3 min, and then the supernatant was removed and the pellet was resuspended in 5 mL pre-warmed YPD medium and incubated at 35 °C or 37 °C (for temperature-sensitive mutants) or 30 °C in a water bath for 1 h. For selective autophagy, 1 OD unit of cells (1 mL culture) was immediately processed for lysis (see below). For non-selective autophagy, the remaining four OD units of cells (4 mL culture) were pelleted at 2500 rpm for 3 min and resuspended in pre-warmed synthetic medium lacking nitrogen (SD-N with 0.17% yeast nitrogen base without ammonium sulphate and amino acids and 2% glucose), incubated for 4 h at 35 °C or 37 °C (for temperature-sensitive mutants) or 30 °C, and then 1 mL of the culture was processed for lysis (see below).

Processing of GFP-Atg8p was performed by monitoring the cleavage of this fusion protein using Western blot analysis. Yeast cultures of 10 mL were grown overnight to an early-to-mid-log phase at OD600 = 0.5–0.8 in uracil drop-out medium at 30 °C. Five OD units of cells were pelleted as above at 2500 rpm for 3 min, the supernatant was removed and the pellet was resuspended in 5 mL pre-warmed uracil drop-out medium and incubated at 30 °C in a water bath for 1 h. After incubation, 1 OD unit of cells (1 mL culture) was immediately processed for lysis (0 h time point) (see below). The remaining four OD units of cells (4 mL culture) were pelleted at 2500 rpm for 3 min, resuspended in pre-warmed synthetic medium lacking nitrogen (SD-N with 0.17% yeast nitrogen base without ammonium sulphate and amino acids, and 2% glucose), incubated at 30 °C, and then 1 mL of culture was taken out after 1 h, 2 h, and 4 h to process for lysis (see below).

### 2.8. Preparation of Yeast Cell Lysates

For pre-Ape1p and GFP-Atg8p processing, 1 mL of culture was transferred immediately on ice and 100% trichloroacetic acid (TCA) was added to 10% final concentration (100 µL), and then the samples were incubated on ice for 20 min. The precipitated proteins were pelleted by centrifugation at 15,000× *g* for 3 min in a microfuge. After washing two times with 1 mL of ice-cold acetone, the pellet was air-dried. The dried pellet was resuspended in 1x Sample Buffer (4x SB recipe: 250 mM Tris-HCl pH 6.8, 40% Glycerol [vol/vol], 8% SDS [w/vol], 0.2% Bromophenol Blue [w/vol]) and then an equal volume of glass beads was added, keeping the glass beads below the level of the liquid. The samples were vortexed for 5 min and incubated at 70 °C for 10 min. Unlysed cells were removed by centrifugation at 10,000× *g* for 1 min in a microfuge and protein extracts equivalent to an OD600 = 0.2 units of cells (20 µL) were loaded on a 12% SDS-polyacrylamide gel (for pre-Ape1p) or a 10% gel (for GFP-Atg8p), resolved by SDS-PAGE, and processed for Western blotting using anti-Ape1 antibody or anti-GFP antibody.

### 2.9. Invertase Secretion Assay

The invertase assay protocol was based on the method described by Munn et al. [64]. Yeast cells were cultured overnight in YPD medium at 24 °C until reaching an OD600 of 0.2–0.5. For each sample, two OD units of cells were collected and washed with 1 mL of distilled water. Subsequently, two OD units of each yeast strain were resuspended in inducing YP(low D)S (0.05% glucose and 2% sucrose) and non-inducing YPD (2% glucose) media to an OD600 of 0.5. The samples were then incubated at 25 °C, 30 °C, and 37 °C for 1 h. Post-incubation, the cells were washed twice with 2 mL of ice-cold 10 mM sodium azide, then adjusted to an OD600 of 0.5 using 10 mM sodium azide. Two 0.5 mL aliquots (0.25 OD units each) were transferred to sterile microfuge tubes: one for whole cell invertase activity (external) and one for lysed cell invertase activity (total; external + internal). For whole cells, 50 µL of distilled water was added, and for lysed cells, 50 µL of 10% Triton-X was added. To lyse the cells, two freeze–thaw cycles were performed using liquid nitrogen and room temperature thawing. The assay was prepared on ice. To 20 µL of cells, 25 µL of 0.2 M sodium acetate (pH 4.9) and 12.5 µL of 0.5 M sucrose were added. The sucrose was added to the tube walls to initiate the reactions simultaneously upon centrifugation. The tubes were centrifuged briefly and placed in a 37 °C water bath for 10 min. The reaction was terminated by adding 50 µL of 100 mM potassium phosphate (pH 7), and invertase was inactivated by heating at 90 °C for 3 min. The samples were chilled on ice. Next, 500 µL of assay mix (50 μg/mL glucose oxidase from *Aspergillus niger*, 10 μg/mL horseradish peroxidase, 10 mM potassium phosphate buffer pH 7, 300 μg/mL o-dianisidine, 38% vol/vol glycerol) was added. The samples were incubated at 30 °C for 20 min. Color development was achieved by adding 750 µL of 6N HCl, and absorbance was measured at A540 using a spectrophotometer.

### 2.10. Western Blot Analysis

Samples (0.2 OD units of cells for yeast lysates and 20 μg total protein for human fibroblast lysates) were fractionated on SDS-polyacrylamide gels (different concentrations depending on the protein analyzed) for 1 h and 30 min at 110 V. The proteins were transferred to a nitrocellulose membrane or PVDF membrane (for LC3-II detection) for 1 h at 100 V in transfer buffer (25 mM Tris, 192 mM glycine, and 20% methanol [vol/vol]). Membranes were blocked with 5% skim milk powder in PBS-T (PBS with 0.1% Tween 20 [vol/vol]) for 2 h and incubated overnight at 4 °C with primary antibodies diluted in PBS-T. The secondary antibodies diluted in PBS-T were added for 1 h at room temperature. After incubation with ECL reagent (Thermo Fisher Scientific, Waltham, MA, USA) for 1 min, membranes were visualized on an Amersham Imager 600 (Chicago, IL, USA).

### 2.11. Genetic Studies

Written informed consent was obtained from the patient’s parents according to an IRB-approved protocol (WCHN). Whole-exome sequencing was performed on genomic DNA obtained from the proband’s blood. The amplified DNA fragments were hybridized to the Agilent SureSelect Human All Exon V4 (51 Mb), the captured library was sequenced on a HiSeq 2000 platform, and the reads were aligned against the human reference genome (GRCh37 at UCSC). Candidate variants were independently verified by Sanger sequencing on genomic DNA samples from blood the patient and her parents. Target loci were amplified by PCR, and PCR products were purified and bidirectionally sequenced using the BigDye Terminator v3.1 Cycle Sequencing Kit (Applied Biosystems, Waltham, MA, USA) on an ABI 3730 DNA Analyzer (Applied Biosystems).

### 2.12. Cell Culture

Primary fibroblasts obtained by skin biopsy from the affected individual were cultivated and propagated using Dulbecco’s modified eagle medium (DMEM) (Wisent) supplemented with 10% (vol/vol) fetal bovine serum (FBS; Thermo Fisher Scientific, Waltham, MA, USA). Cells were then incubated at 37 °C in a humidified incubator with 5% CO_2_.

### 2.13. Autophagy Assays in Mammalian Cells

Primary fibroblasts seeded in 6 cm diameter dishes were washed twice with phosphate-buffered saline (PBS) and incubated with Earl’s balanced salt solution (EBSS) (Wisent) for 0 h, 1 h, and 2 h to induce autophagy. The cells were then lysed using 150 µL lysis buffer (150 mM NaCl, 50 mM Tris pH 7.2, 1 mM DTT, 1% Triton X-100, 0.5 mM EDTA, complete protease inhibitors (Roche Diagnostics, Basel, Switzerland)) and analyzed by Western blotting for the processing of LC3. The data were normalized to the tubulin signal developed from the same membrane as the LC3 blot.

### 2.14. Antibodies and Dyes

The antibodies used were rabbit anti-Ape1 (1:2500) (gift from Dr. Daniel Klionsky, University of Michigan), mouse anti-GFP (1:1000) (Roche Diagnostics, Basel, Switzerland), rabbit anti-LC3 (1:2500) (Abcam, Cambridge, UK), mouse anti-tubulin (1:10,000) (Sigma-Aldrich, Burlington, MA, USA), rabbit anti-mCherry (1:4000) (Abnova, Taipei City, Taiwan), rabbit anti-mannosidase II (1:500) (gift from Dr. Kelly Moreman, University of Georgia). To visualize the DNA, staining with Hoechst 33342 (1:2000) (Invitrogen, Carlsbad, CA, USA) was used. Except for anti-mannosidase II and Hoechst, which were used for immunofluorescence microscopy, all the other antibodies were used for Western blotting. The secondary antibodies used in this study for Western blot analysis were goat anti-Rabbit (1:5000) (Seracare, Milford, MA, USA) and goat anti-Mouse (1:5000) (Seracare, Milford, MA, USA), while for immunofluorescence anti-Rabbit Alexafluor647 (Life Technologies, Carlsbad, CA, USA) was used.

### 2.15. Immunofluorescence Microscopy

Fibroblast cells cultured in 12 or 6-well plates on coverslips were washed twice with PBS, fixed for 15 min at room temperature with 4% paraformaldehyde (PFA), quenched for 10 min in 0.1 M glycine, and then permeabilized for 7 min in 0.1% Triton X-100 in PBS. The cells were blocked with 5% blocking solution (5% normal goat serum in PBS) for 45 min at room temperature and then incubated with primary antibodies resuspended in 5% blocking solution overnight at 4 °C. The cells were washed twice with PBS for 10 min each time and secondary antibodies resuspended in 5% blocking solution were added for 45 min at room temperature. The coverslips were washed one time with Hoescht in PBS for 2 min and two other times with PBS for 10 min each wash. The coverslips were then dried for 2–3 min and mounted on clean slides using Prolong Gold AntiFade (Life Technologies). Nikon confocal laser scanning microscope C2 TIRF was used to take images of the cells. The Golgi fragmentation images were obtained with 0.20 µm increment size and ImageJ was used to delineate the cells for each condition. The delineated cell images were converted to IMS files and imported to Imaris, and then analyzed using parameters pre-set to recognize Golgi structures as spherical-shaped structures (fragments) no smaller than 0.550 µm in size. The number of fragments were identified by the software and the number of fragments was quantified. The N value for control cells was 75 and the N values for S1 (patient cells) and S1 + RFP-TRAPPC1 (rescued cells) were 78 and 81, respectively.

### 2.16. Retention Using Selective Hooks (RUSH) Assay

Fibroblasts cultured in DMEM supplemented with 10% fetal bovine serum were used to monitor ER-to-Golgi trafficking through the Retention Using Selective Hooks (RUSH) assay as previously described [65]. ImageJ software, version 1.54, was used to quantify the fluorescence intensity in individual cells. First, the fluorescence intensity of the Golgi was measured in each cell to be quantified. For this, the Golgi was distinguished and outlined manually in every cell and the fluorescence intensity was measured for each time point in the movie (images were captured every two minutes). With the same Golgi outlining, a random spot was chosen as the background and this background value was used to correct the Golgi fluorescence intensity at each time point. Next, the total fluorescence intensity was measured for each time point by outlining the entire cell. With the same cell outlining, a “background” measurement was taken. For each time point, the background-corrected Golgi fluorescence intensity was divided by the background-corrected total cell fluorescence intensity. The produced values (the ratio of Golgi fluorescence intensity/total cell fluorescence intensity) were plotted.

## 3. Results

### 3.1. TRAPPC1, TRAPPC2, TRAPPC2L, TRAPPC6A, and TRAPPC6B Successfully Replace Their Yeast Orthologs BET5, TRS20, TCA17, and TRS33, Respectively

In an effort to create a platform in yeast that can be used for characterizing mutations in TRAPP subunits in the absence of patient-derived cells, and since many reported mutations affect the core subunits, we started humanizing the core of the TRAPP complex. The human TRAPP genes and proteins TRAPPC1, TRAPPC2, TRAPPC2L, TRAPPC3, TRAPPC4, TRAPPC5, TRAPPC6A, and TRAPPC6B, will be referred to as C1, C2, C2L, C3, C4, C5, C6A, and C6B, respectively. These proteins are part of the core in both yeast and humans, common to both TRAPP complexes, except for C2L (Tca17p ortholog), which is only found in the TRAPP II complex in yeast [15]. Each human open reading frame (ORF) was PCR amplified as a repair template with flanking homology to the locus of interest in yeast and was assessed by replacing its yeast ortholog using CRISPR/Cas9 as an editing tool. Eight human TRAPP genes (*C1*, *C2*, *C2L*, *C3*, *C4*, *C5*, *C6A*, and *C6B*) were tested for their ability to complement their yeast orthologs (*BET5*, *TRS20*, *TCA17*, *BET3*, *TRS23*, *TRS31*, and *TRS33*, respectively) individually. As seen in Figure 1A, after the transformation of the yeast cells, reactions containing *C1*, *C2*, *C2L*, *C6A*, and *C6B* repair templates showed colony growth in the selective media, suggesting that replacement has taken place, while those containing *C3*, *C4*, and *C5* did not show any colonies.

To confirm the success of replacement for *C1*, *C2*, *C2L*, *C6A*, and *C6B* genes, colony PCR was performed using confirmation primers designed for each of the transformed loci in yeast. As shown in Figure 1B, all the replaced loci in yeast show a band of the expected size, suggesting successful integration of the repair templates.

For further verification and to confirm a scar-less replacement of the yeast gene with the human ORF, the respective humanized loci were subjected to Sanger sequencing and all the sequences confirmed the presence of human ORFs in their respective yeast loci, replacing only the sequence between the start and stop codons and not affecting the 5’ or 3’ untranslated regions (Appendix A). The results from these experiments suggest that *C1*, *C2*, *C2L*, *C6A*, and *C6B* are able to complement their yeast orthologs while complementation was not successful for *C3*, *C4*, and *C5*.

### 3.2. TRAPPC3, TRAPPC4, and TRAPPC5 Cannot Complement the Loss of BET3, TRS23, and TRS31, Respectively, in Yeast

Since the humanized yeast strains express the human gene from the endogenous yeast promotor, the reason for the unsuccessful replacements of *C3*, *C4*, and *C5* might be due to their inability to compensate for the loss of the genes encoding their essential yeast orthologs *BET3*, *TRS23*, and *TRS31*, respectively, at endogenous levels. We therefore examined if they could compensate when overexpressed. Haploid yeast strains in which the genomic copies of *BET3*, *TRS23*, and *TRS31* were disrupted *(bet3*∆, *trs23*∆, *trs31*∆) but the cells were kept alive with *URA3*-based plasmids containing *BET3*, *TRS23*, or *TRS31*. These plasmids can be counter-selected on plates that contain 5-fluoroortic acid (5-FOA). The *bet3*∆, *trs23*∆, and *trs31*∆ strains were transformed with empty vectors, or with vectors containing either *BET3* (the *bet3*∆ strain), *TRS23* (the *trs23*∆ strain), or *TRS31* (the *trs31*∆ strain), or *C3* (the *bet3*∆ strain), *C4* (the *trs23*∆ strain), or *C5* (the *trs31*∆ strain), under the control of different overexpression promoters (GPD or ADH1). As shown in Figure 2, none of the human TRAPP genes could compensate for the loss of *BET3*, *TRS23*, and *TRS31* as indicated by the lack of growth on plates containing 5-FOA. These results indicate that *C3*, *C4*, and *C5* are not able to compensate for the disruption of their orthologous genes in yeast and reinforce the previous results from the humanization experiments in yeast where *C3*, *C4*, and *C5* were shown to be incapable of replacing their yeast counterparts *BET3*, *TRS23*, and *TRS31*, respectively.

### 3.3. Humanized Yeast Strains Show a Slower Growth Phenotype Compared to the Parental Yeast Strains

A previous study with humanized yeast strains revealed that although most yeast strains might be slightly affected by the process of humanization, many other humanized strains show highly reduced growth rates that can be up to 70% slower [66]. Therefore, we sought to test the growth rate of the humanized yeast strains with *C1*, *C2*, *C2L*, *C6A*, and *C6B* genes. For this, serial dilutions of these humanized strains were spotted onto YPD plates. The plates were incubated at three different temperatures (30 °C, 35 °C, 37 °C) and, after 48 h, examined for growth. As shown in Figure 3A, yeast strains humanized with *C1*, *C2*, and *C2L* showed a slower growth phenotype compared to the parental strain (MSY135) at 30 °C, 35 °C, and 37 °C, which is also the optimal temperature for human genes to function. In contrast, yeast strains humanized with *C6A* and *C6B* genes appeared to be unaffected by any of the conditions.

Next, we quantitatively tested the growth rate of the humanized yeast strains in a liquid medium. As shown in Figure 3B, *C1*, *C2*, and *C2L* strains showed slower growth rates compared to the parental strain in all the temperatures with a 2–5 h lag phase. *C6A* and *C6B* strains showed growth rates similar to the parental strain at 30 °C and 35 °C, while at 37 °C, the growth rate of the *C6B* strain was slower compared to the parental strain. This was in contradiction to the previous experiments on solid plates where the *C6B* strain appeared to grow normally even at 37 °C. These experiments demonstrate that humanized yeast strains with *C1*, *C2*, and *C2L* overall show a partial reduction in fitness compared to the parental strain, while the growth of humanized yeast strains with *C6A* and *C6B* is not affected at low temperatures and show similar fitness to the parental strain, but *C6B* appears to be heat-sensitive.

### 3.4. A Novel Disorder Associated with Biallelic TRAPPC1 Variants

The proband who carries these biallelic *C1* variants is an 11-year-old girl with a severe neurodevelopmental phenotype. She was born to healthy and unrelated parents following an uncomplicated at-term pregnancy by normal delivery. She then presented with global neurodevelopmental delay and a phenotype characterized by truncal hypotonia with lower limb spasticity, intellectual disability with severely impaired adaptive functioning, lack of speech, and autistic features with self-injurious behavior. She has some dysmorphic features including hypertelorism, long palpebral fissures, high-arched eyebrows, a bulbous nasal tip, large ears, a high-arched palate, widely spaced teeth, and a single palmar crease on the right hand. Brain MRI revealed a thin corpus callosum, hypoplastic olfactory nerves, and hypomyelination. She also has a myopathy with persistently elevated creatine kinase (CK) levels. A muscle biopsy showed minor non-specific findings and no ultrastructural anomalies.

Whole exome sequencing, initially reported as negative, was reanalyzed through the Program for Undiagnosed Rare Diseases of CIBERER (ENoD) [67]. Two candidate variants were identified in the *TRAPPC1* gene: NM_021210.5: c.362_363del; p.(Val121Alafs*3)/NM_021210.5: (c.64_72del); p.(His22_Lys24del). Ulterior validation by Sanger sequencing and segregation in parental samples revealed that the patient is a compound heterozygote with biallelic variants in *C1*, while both parents are heterozygous carriers. In the maternally inherited variant, p.(Val121Alafs*3), a deletion of two nucleotides (c.362_363del) results in a frameshift mutation that introduces a premature stop codon, altering the C-terminal 24 amino acids that delete one of the three helices (Figure 4A,B). This variant (rs768379826) is rare (allele population frequency in the gnomAD 4.0 database of 0.0002763, with no homozygotes) and has a very high in silico prediction of deleteriousness (CADD value of 33). In the paternally inherited variant, p.(His22_Lys24del), a deletion of nine nucleotides (c.64_72del) results in an in-frame three amino acid deletion in the protein in a short loop connecting two strands of the β-sheet (Figure 4A,B). This variant is very uncommon (allele population frequency in the gnomAD 4.0 database of 0.0000012) and has a high in silico prediction of deleteriousness (CADD value of 22.4).

### 3.5. Humanized Yeast as a Platform for Characterizing the First Reported Mutation in TRAPPC1

We next sought to test whether the humanized platform in yeast provides an optimal system for characterizing mutations in human genes. For this, we studied the first reported mutation in *C1*. *C1* is an essential gene in humans, encoding for a protein ortholog to Bet5p in yeast that shares 33% identity and 54% similarity (Table 1). C1 is part of the longin domain proteins, which are composed of a conserved domain of 120–140 amino acids at their N-terminus [68]. Longin domains contain a five-stranded β-sheet flanked by a single α-helix on one side and two α-helices on the other side [69], and they have been proposed to regulate many membrane-trafficking processes [70]. C1 and Bet5p are part of the core TRAPP subunits and, therefore, they are common to both TRAPP II and III.

There were two reasons for using the mutant *C1* variants as a candidate to test our humanized platform in yeast. First was the fact that wild-type *C1* can replace *BET5* in yeast, and second was the presence of fibroblast cells from the patient bearing the *C1* variants. This would allow us to compare the phenotype of the humanized yeast platform with that of human cells bearing these variants. As a source of each variant for our yeast work, messenger RNA (mRNA) was extracted from the patient fibroblasts and used to synthesize complementary DNA (cDNA). This cDNA was used as a template to generate the repair templates to humanize the yeast for both *C1* mutations.

As a first estimate to see whether we would be able to humanize yeast strains with both *C1* mutations, we examined the overexpression of each variant in yeast devoid of *BET5* (*bet5*Δ) containing a *URA3*-based wild-type *BET5* in the presence of 5-FOA. As shown in Figure 5A, only *BET5*, *C1*, and *C1 (c.64_72del)* could compensate for the loss of *BET5* as indicated by growth on plates containing 5-FOA, while *C1 (c.362_363del)* was non-functional and did not support the growth of *bet5*∆ yeast strains on 5-FOA.

Knowing that when overexpressed, the maternal variant C1 p.(Val121Alafs*3) (expressed from *C1 (c.362_363del)*) is lethal and the paternal variant C1 p.(His22_Lys24del) (expressed from *C1 (c.64_72del)*) is functional, we examined their functionality in the humanized yeast platform under the control of the endogenous *BET5* promoter. For this, we used the CRISPR/Cas9 plasmid and either *C1 (c.362_363del)* or *C1 (c.64_72del)* as repair templates. As seen in Figure 5B, after the transformation of the yeast cells, humanized strains with *C1* and *C1 (c.64_72del)* showed colony growth in the selective media while humanized strains with *C1 (c.362_363del)* could not support yeast growth.

C1 p.(Val121Alafs*3) (expressed from *C1 c.362_363del*) was lethal and suggested that the paternal variant C1 p.(His22_Lys24del) (expressed from *C1 c.64_72del*) was functional at endogenous Bet5p levels. Replacement with either wild-type *C1* or *C1 (c.64_72del)* was confirmed by colony PCR (Figure 5C, upper panel) and DNA sequencing (Figure 5D, lower panel). These results confirm the functionality of *C1 (c.64_72del)* in yeast and the ability of this mutant to complement the *BET5* gene.

### 3.6. Humanized Yeast with C1 (c.64_72del) Is Conditional-Lethal

Since the *C1 (c.64_72del)* mutant was functional and could replace the *BET5* gene in yeast, we next examined the growth efficiency of this mutant in solid and liquid media using the parental strain (MSY135) and the humanized *C1* strain as controls. As shown in Figure 6A, yeast strains humanized with *C1 (c.64_72del)* showed a slower growth phenotype compared to the parental strain (MSY135) at 30 °C, even slower growth at 35 °C, and no growth at 37 °C. These observations are consistent with the results from the liquid growth curves (Figure 6B) where the *C1 (c.64_72del)* strain showed a slower growth rate at 30 °C compared to the parental MSY135 strain and was essentially non-functional at 35 °C and 37 °C. These results demonstrate that the *C1 (c.64_72del)* humanized yeast strain is a conditionally lethal mutant, and its growth is sensitive to temperatures above 30 °C. Although the *C1 (c.64_72del)* mutation complements *BET5* in yeast, its functionality is not comparable to the wild-type *C1* at 30 °C.

### 3.7. Non-Selective Autophagy Is Defective in the Humanized C1 (c.64_72del) Yeast Strain

The fact that the mutant *C1 (c.64_72del)* was conditionally lethal presented us with the ability to characterize this mutation in our humanized platform by growing this strain at permissive and restrictive temperatures. Since the C1 subunit is part of both TRAPP complexes in humans, and knowing that TRAPP III functions in autophagy, we tested whether *C1 (c.64_72del)* affected autophagy.

Depending on the media in which the cells are grown, the presence of a nitrogen source (rich media) or the absence of a nitrogen source (starvation media) can help distinguish between the selective cytosol-to-vacuole (Cvt) autophagic pathway (+nitrogen) and the non-selective pathway (- nitrogen). The Cvt pathway occurs continuously during cell growth in conditions rich in nutrients, while the non-selective pathway is usually triggered by starvation [71]. The marker protein Ape1p is delivered to the vacuole by the selective pathway in the presence of nitrogen, and in the absence of nitrogen it is transported to the vacuole in a non-selective manner [72]. The processing of Ape1p is monitored by detecting the levels of the precursor protein and its processed form by SDS-PAGE/Western blot analysis. As shown in Figure 7A, the processing of Ape1p in the presence of nitrogen (+N panel) was completely blocked for the humanized strain with *C1* and *C1 (c.64_72del)* compared to the parental strain, suggesting that selective autophagy is impaired in the two humanized strains. In starvation conditions, when Ape1p is processed by non-selective autophagy (−N panel), at 30 °C, there was a notable amount of precursor Ape1p and a very small amount of mature Ape1p detected in the humanized *C1 (c.64_72del)* strain compared to the humanized *C1* strain and the parental strain, suggesting a defect in non-selective autophagy of Ape1p in the *C1 (c.64_72del)* strain (Figure 7A). At 35 °C, no processing of the Ape1p was detected from the precursor to the mature form in the humanized *C1 (c.64_72del)* strain compared to the humanized *C1* strain and the parental strain, although the humanized *C1* strain at this temperature showed less processing of Ape1 compared to the parental strain. At 37 °C, both humanized strains with *C1* and *C1 (c.64_72del)* were defective in non-selective autophagy and did not process Ape1p compared to the parental strain. These results indicate that humanized strains with *C1* and *C1 (c.64_72del)* are both defective in selective autophagy while *C1 (c.64_72del)* impairs the non-selective autophagy of Ape1p.

To further evaluate whether the *C1 (c.64_72del)* mutation affects non-selective autophagy, we examined the processing of GFP-Atg8p. Under starvation conditions, this fusion protein is delivered to the vacuole in a non-selective manner and cleaved, releasing GFP, which is resistant to the proteolytic enzymes of the vacuole. The appearance of free GFP can be used as a marker, which can be detected through Western blot analysis over time. As seen in Figure 7B, at 30 °C, in the parental strain and the humanized *C1* strain, there was an increase in the amount of free GFP over time, whereas the humanized *C1 (c.64_72del)* strain accumulated only the full-length GFP-Atg8p but not free GFP. Collectively, these results demonstrate that *C1 (c.64_72del)* affects the non-selective autophagy of Ape1p and GFP-Atg8p while both the mutant and wild-type *C1* appear to affect selective autophagy.

### 3.8. Secretion Is Defective in the Humanized C1 (c.64_72del) Yeast Strain

Since TRAPP III also plays a role in ER-to-Golgi trafficking and TRAPP II functions at the level of the Golgi, a defect in C1 would be expected to affect secretion. We tested this notion by examining the secretion of invertase, the product of the *SUC2* gene. In high (2%) glucose, the gene is not expressed, but expression is induced in low (0.05%) glucose in the presence of the substrate sucrose [64]. Yeast cells were grown under non-inducing and inducing conditions at various temperatures. We included a *sec18* mutant, which served as a control for a secretion block. Invertase was assayed on whole cells (a measure of external invertase activity) and on lysed cells (a measure of total invertase activity). We then calculated the percent secreted invertase and internal invertase. As shown in Figure 7C,D, wild-type cells efficiently secreted invertase upon induction with no significant change seen at any of the temperatures examined. As expected, the percentage of invertase that was secreted in *sec18* was comparable to that of control yeast cells at 25 °C but dropped precipitously as the growth temperature increased to 34 °C, with a concomitant increase in internal invertase. The *C1* humanized yeast showed no significant difference in secreted invertase when compared to the wild type at any of the temperatures. In contrast, the *C1 (c.64_72del)* humanized strain showed a significant defect in secretion at 25 °C that was more significantly seen at 34 °C. These results indicate that the *C1 (c.64_72del)* mutant affects secretion and does so in a temperature-sensitive manner.

### 3.9. Fibroblasts from the Affected Individual with TRAPPC1 Mutations Do Not Show an Accumulation of LC3-II during Starvation

To better establish the humanized yeast system as a valid system to use in the absence of human cells, we needed to compare the data obtained from the humanized yeast to fibroblasts obtained from the affected individual with the compound heterozygous *TRAPPC1* variants. We therefore characterized these cells for an autophagic defect to assess whether the results from this experiment would parallel those in yeast. For this purpose, we starved the fibroblasts and monitored the levels of LC3-II. LC3, the mammalian ortholog of yeast Atg8p, is synthesized as pro-LC3, a soluble unprocessed form that is proteolytically processed into LC3-I and finally modified into LC3-II (the phosphatidylethanolamine-conjugated form), which is associated with membranes [73].

Control and patient fibroblasts were starved for up to 2 h in EBSS starvation medium to induce autophagy. Cell lysates were collected at 0 h, 1 h, and 2 h time points. As indicated in Figure 8A and quantified in Figure 8B, control cells showed an increase of LC3-II over the 2 h period. In contrast, patient cells did not show significant differences in the level of LC3-II over time compared to the control cells, while they showed stable levels of LC3-I over the same 2 h period. The failure to convert LC3-I to LC3-II suggests a defect in autophagosome formation in these cells.

### 3.10. Wild-Type TRAPPC1 Rescues a Membrane Trafficking Defect and Golgi Fragmentation in Patient Fibroblasts

Human TRAPP II and III are involved in ER-to-Golgi trafficking and late Golgi transport, respectively, and the C1 protein is part of the core TRAPP subunits of both complexes. Hence, patient fibroblasts were examined for defects in membrane trafficking using the RUSH system. In this assay, a fluorescent cargo protein (ST-eGFP) is kept in the ER until biotin is administered. Upon biotin addition, the cargo exits the ER and is transported to the Golgi, and the fluorescent signal is monitored and quantified as it accumulates in the Golgi [44,65].

As shown in Figure 9A, the perinuclear fluorescence intensity in the control cells increased over time as the cargo protein arrived at the Golgi. In contrast, the patient fibroblasts (S1) displayed a delay in the arrival of the cargo protein at the Golgi compared to the control cells. These results demonstrate that patient fibroblasts are defective in membrane-trafficking events along the biosynthetic pathway. This defect is due to non-functional *C1* since patient cells transfected with a construct of the wild-type *C1* tagged with a red fluorescent protein (RFP) rescued the trafficking defect (Figure 9A).

Next, we examined the Golgi for alterations. Previous studies have reported changes in Golgi morphology in patients with mutations in TRAPP subunits [39,43,44]. With this in mind, the fragmentation of the Golgi into small structures was quantified in the patient fibroblasts (S1) and was compared to the control cells. To visualize the Golgi, an antibody that recognizes the Golgi resident protein mannosidase-II (Man-II) was used. As shown in Figure 9B, the number of Golgi fragments in patient cells was significantly higher compared to the control. The transfection of patient cells with a construct of the wild-type *C1* fused to RFP significantly rescued this phenotype, confirming that the Golgi morphological alterations are *C1*-dependent.

## 4. Discussion

The scarcity of human cellular models represents a major challenge for studying the globally increasing number of rare diseases in general and TRAPPopathies in particular. This can be attributed at least in part to the limited number of patients diagnosed with rare conditions who agree to provide cells through invasive techniques such as skin and deeper tissue biopsies. To overcome this difficulty, there is a need for using other models to characterize disease-causing variants in humans. The yeast *Saccharomyces cerevisiae* has long been used as a model organism for studying human processes due to the remarkable degree of gene conservation between the two organisms. Therefore, using CRISPR/Cas9 as an editing tool, we generated a platform in yeast that would enable the characterization of mutations in human TRAPP genes in the absence of patient-derived cells. Analyzing first the core TRAPP genes, we demonstrated that *C1*, *C2*, *C2L*, *C6A*, and *C6B* can complement individually their orthologs *BET5*, *TRS20*, *TCA17*, and *TRS33*, respectively, in yeast. However, *C3*, *C4*, and *C5* could not replace their corresponding yeast counterparts, *BET3*, *TRS23*, and *TRS31*, respectively. These results were confirmed by plasmid-based assays in yeast, which proved that none of these genes, even when expressed under strong promoters, were able to compensate for the loss of their yeast orthologs.

Despite the successful and scar-less replacement of their yeast equivalents by some of the human core TRAPP genes, *C1*, *C2*, and *C2L* humanized strains revealed slightly compromised fitness compared to the parental yeast strain, a feature more pronounced at temperatures above 30 °C, while the *C6B* humanized strain was found to be heat-sensitive. The growth delays observed in the humanized strains here have also been reported in other studies that explored humanized yeast systems [66,74]. Although the exact mechanisms that contribute to this decrease in growth efficiency are not known, the adverse environment in yeast, alterations in function and regulation of human genes, or failure of human orthologs to complement the moonlighting functions of a yeast protein might be contributing factors.

After generating a set of individually engineered strains with human TRAPP genes expressed at their native loci, the applicability of this system in studying human variants was investigated. Within the context of this study, the focus was to use this system as a test platform to characterize the first reported compound heterozygous mutations in *C1*, where deletion of two nucleotides (c.362_363del) in the maternal variant (C1 (p.Val121Alafs*3)) alters an entire α-helix at the C-terminus of this longin domain protein, while deletion of nine nucleotides (c.64_72del) in the paternal variant (C1 p.(His22_Lys24del)) affects a short loop bridging two strands of the β-sheet structure. Towards this goal, yeast humanization with both mutants individually showed that the *C1 (c.362_363del)* variant is lethal and cannot replace *BET5* in yeast, while the *C1 (c.64_72del)* variant is functional and can complement its ortholog *BET5*. The *C1 (c.64_72del)* variant was tested for its growth efficiency under different temperature conditions (30 °C, 35 °C, and 37 °C) and the growth rate of this mutant was substantially compromised at 35 °C and 37 °C, indicating that it is a conditionally lethal mutant.

The phenotypes observed for these two mutants might be better explained by looking at the structure of the C1 protein and the data that come from crystallographic and cryo-EM studies that have revealed the structure of the TRAPP core. The octameric core forms an elongated rod with two flattened surfaces that accommodate the C1 and C4 proteins in the center. As two longin domain proteins, C1 and C4 also form most of the catalytic site for activating Rab GTPases. As shown in Figure 4B, C1 has three close core interactors, C4 (in light pink), C3 (in cyan), and C6 (in hot pink). In the C1 (p.Val121Alafs*3) variant (expressed from *C1 c.362_363del*), the entire α-helix at the C-terminus of the protein, which is in close proximity to the C6 subunit, is deleted. Therefore, the removal of this α-helix would be expected to have a detrimental effect on C1 as it would interfere with the structure of this longin domain protein. This might explain the non-functionality of the maternal variant C1 p.(Val121Alafs*3) in the yeast. On the other hand, in the C1 p.(His22_Lys24del) variant (expressed from *C1 c.64_72del*), the three amino acid deletion affects a portion of the protein, which is a short peptide that connects two domains of secondary structure and is close to the catalytic site where C1 and C4 accommodate and activate Rab GTPases. Although previous reports have revealed the importance of loops in the folding/geometry of proteins, the partial truncation of this loop might reflect less severe effects on the geometry of C1. Indeed, previous studies have revealed that when perturbations are introduced by mutations in the structure of the protein, they might change the connection between elements of the system (atoms/residues), which in turn might provoke changes in the functional site and other sites of the structure [75,76,77]. This might explain why in fact the paternal variant C1 p.(His22_Lys24del), contrary to C1 p.(Val121Alafs*3), acts as a conditionally lethal variant.

Taken together, our data highlight the efficiency of the humanized yeast as a platform to characterize mutations in human TRAPP genes and suggest a possible correlation between the C1 variant and the severe neurodevelopmental delay from which the individual bearing the *C1* mutations suffers. This latter suggestion is related to the use of this system for addressing VUS. VUS are variants for which the clinical significance, and therefore the association with the risk of disease, is not clear [9]. In this study, we applied this platform for addressing the significance of a compound heterozygous mutation in *TRAPPC1* identified as a VUS from genetic testing data. While the maternal variant was found to be lethal and could not replace its yeast counterpart, the paternal variant manifested as a conditional phenotype. Therefore, employing this system with other TRAPP proteins will allow us to rapidly test additional VUS and determine if they are functionally significant, thereby accelerating the diagnostic odyssey for rare disease patients.

Although we tried to detect the expressed TRAPPC1 protein in yeast lysates from the humanized strains, the commercial antibody we used did not work well by Western blotting and the results were ambiguous. We suspect that the wild-type protein is indeed expressed since the yeast homolog Bet5p is essential and the TRAPP core would be predicted to not assemble in its absence based on the known structures of the complexes from multiple organisms [78,79,80,81]. It is possible that the conditional phenotype of the paternal variant is unstable at elevated temperatures in both yeast and human cells, possibly accounting for the human disorder.

It was surprising that several yeast genes could not be functionally replaced by their human orthologs, particularly *BET3*, which shows 56% identity and 76% similarity at the protein level with human C3. It is important to note that C3 is the only core protein that is present in two copies (C3a and C3b) by flanking both sides of the core center formed by C1 and C4. This might explain why its humanization is intolerable since the single gene replacement should generate in a sense a double-protein replacement in the complex. Therefore, a major challenge in the future will be to make these genes humanizable in yeast. Since certain genes require local interactions for functional replaceability, we are exploring combinations of replacements. As we progressively humanize additional TRAPP genes in yeast, our final goal would be to assemble all these replacements in one strain, generating in this way a fully humanized TRAPP core in yeast.

Through the humanized system, we were able to characterize the first reported mutations in *C1* by performing functional experiments that explore pathways where this protein is involved. The results from these experiments were supported by similar outcomes in patient fibroblasts confirming the applicability of this system as a reliable tool for characterizing other gene mutations. Since the *C1* mutation in this study is compound heterozygous, it would be of interest to test this mutation in diploid yeasts with one allele as the maternal copy and the other allele as the paternal copy. This may reveal additional phenotypes that the single mutations might mask.

It is noteworthy that defects in both ER-to-Golgi traffic and autophagy have been linked to neurodevelopmental diseases [82,83]. Though the disease mechanisms are not well understood, our humanized yeast system showed defects in both processes, as did the human fibroblasts, further validating the utility of such a system to study complex human diseases.

In conclusion, this study, together with many others before, highlights the crucial importance of using yeast as a model organism for exploring human genetics and disease in a simplified background. The engineered strains in this study will be further used as effective platforms to characterize other disease-causing mutations.

## Figures and Tables

**Figure 1 cells-13-01457-f001:**
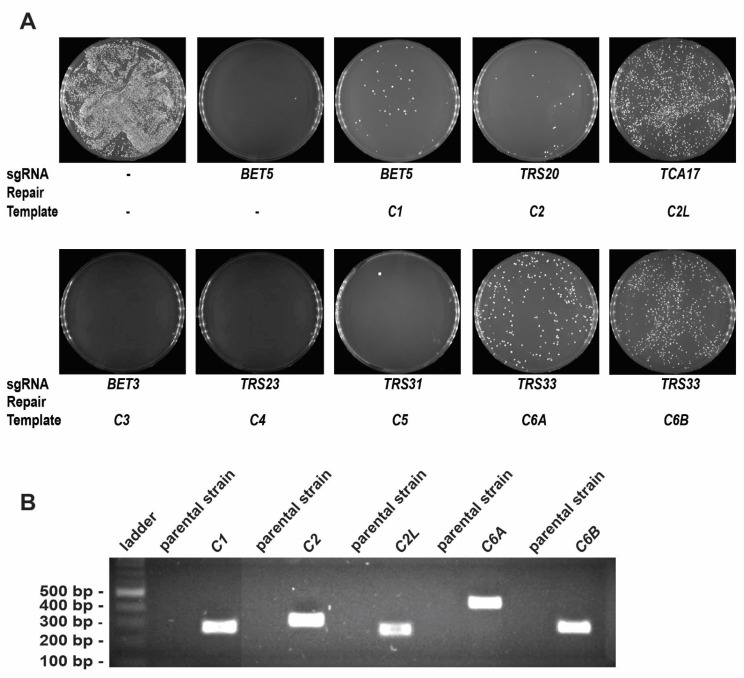
Yeast humanization with core TRAPP genes using a CRISPR/Cas9 editing tool. (**A**) Yeast cells were transformed with ORFs from human *C1*, *C2*, *C2L*, *C3*, *C4*, *C5*, *C6A*, or *C6B* as repair templates in the presence of the self-containing CRISPR/Cas9 plasmid to replace their yeast orthologs—*BET5*, *TRS20*, *TCA17*, *BET3*, *TRS23*, *TRS31*, and *TRS33*, respectively. The upper left plate is a control of cells transformed with the empty CRISPR/Cas9 plasmid that contains no sgRNA, and no repair template is provided; therefore, it is used to evaluate the transformation efficiency of the yeast strains being used. The remaining plates show the results of the indicated sgRNA with or without (second plate from left at top as an example of the double-stranded break (DSB)-induced lethality) repair template. The presence or absence of growth in the remaining plates is an indication of rescue or no rescue from the DSB lethality of the Cas9 nuclease when an appropriate repair template is provided. (**B**) Colonies from plates transformed with the empty CRISPR/Cas9 plasmid and from plates transformed with *C1*, *C2*, *C2L*, *C6A*, or *C6B* repair templates were picked and analyzed by PCR for correct integration of the human ORF. Lanes labeled parental strain represent negative controls from the parental strain transformed with the empty plasmid. The remaining lanes represent the lysates from cells transformed with the appropriate repair templates. The expected band size for each successful transformation is 274 bp for *C1*, 311 bp for *C2*, 255 bp for *C2L*, 428 bp for *C6A*, and 276 bp for *C6B*.

**Figure 2 cells-13-01457-f002:**
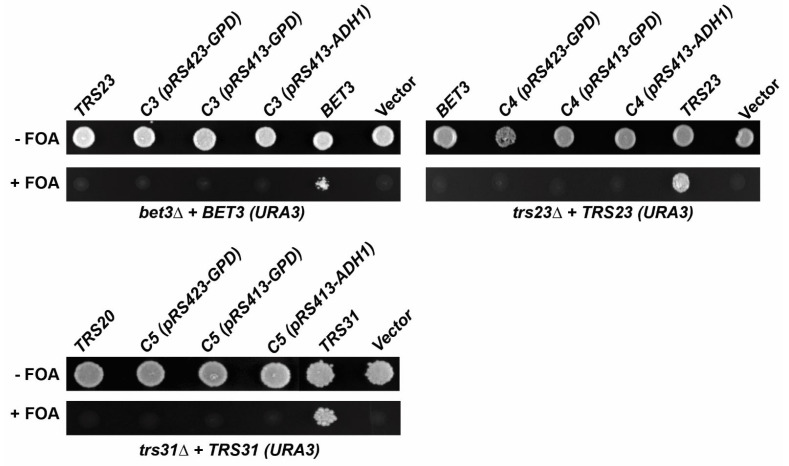
*C3*, *C4*, and *C5* cannot compensate for the loss of *BET3*, *TRS23*, and *TRS31* in yeast. Haploid yeast strains *bet3*∆, *trs23*∆, and *trs31*∆ were kept alive with *URA3*-based *BET3*, *TRS23*, and *TRS31* plasmids, respectively. These strains were transformed with empty plasmids (vector), or vectors containing either *BET3* (the *bet3*∆ strain), *TRS23* (the *trs23*∆ strain), or *TRS31* (the *trs31*∆ strain) under their endogenous promoters or either *C3* (the *bet3*∆ strain), *C4* (the *trs23*∆ strain), or *C5* (the *trs31*∆ strain) under the control of the *ADH1* or *GPD* promoters. Transformed cells were plated on 5-FOA to counter-select for the *URA3*-based plasmids.

**Figure 3 cells-13-01457-f003:**
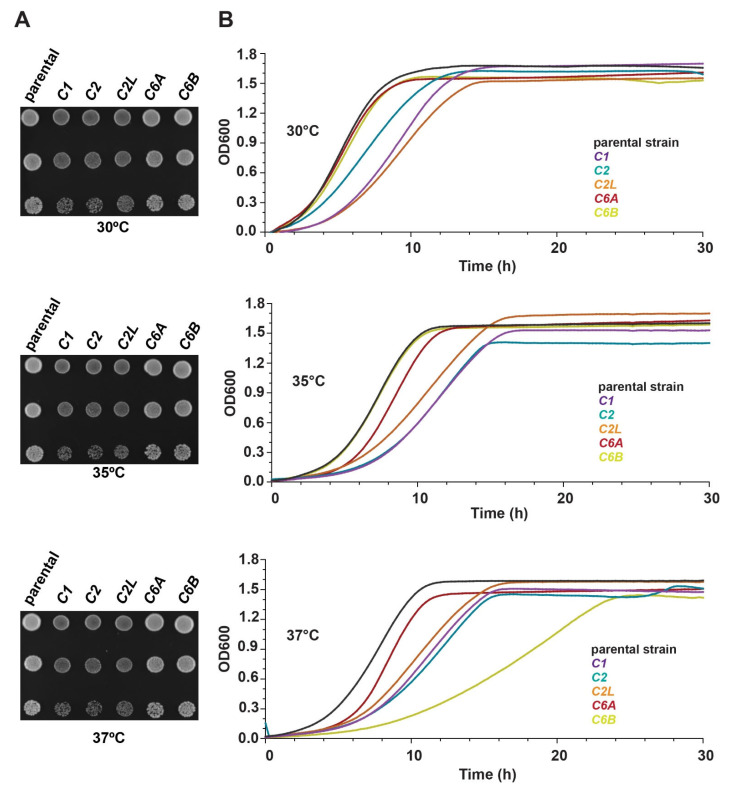
Growth efficiency of the humanized yeast strains in solid and liquid media. (**A**) Images from the plates of *C1*, *C2*, *C2L*, *C6A*, and *C6B* humanized yeast strains. Serial dilutions of the liquid cultures were spotted on YPD plates and incubated for 48 h at three different temperatures, 30 °C, 35 °C, and 37°C. (**B**) Representation of the growth curves in liquid medium for *C1*, *C2*, *C2L*, *C6A*, and *C6B* humanized yeast strains in 30 °C, 35 °C, and 37 °C. Shown are the averages of OD600 for three biological replicates.

**Figure 4 cells-13-01457-f004:**
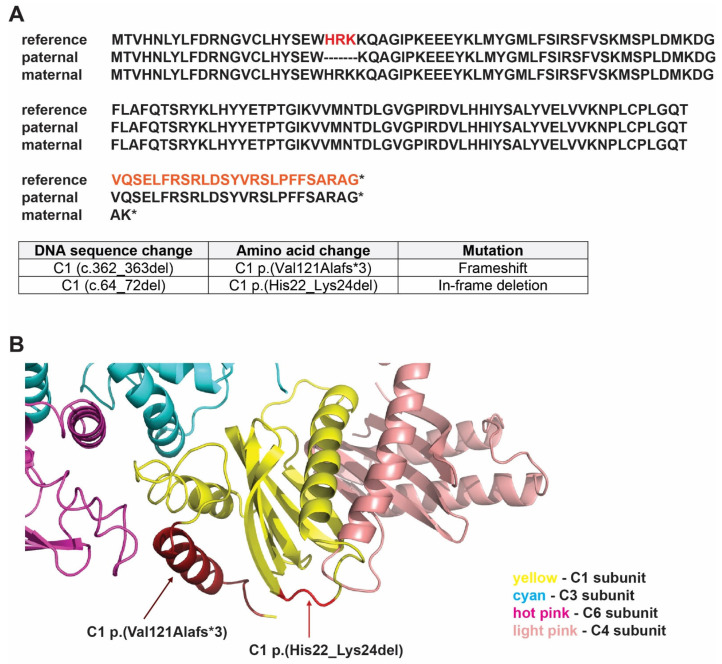
The compound heterozygous mutation in the C1 protein. (**A**) C1 p.(Val121Alafs*3) and C1 p.(His22_Lys24del) variants are predicted in the amino acid sequence of the C1 protein. Shown are the reference and the predicted sequence of the protein for both variants and the nomenclature for *C1* mutations and variants. (**B**) C1 p.(Val121Alafs*3) and C1 p.(His22_Lys24del) are highlighted in red in the structure of the C1 protein. Interacting subunits of the TRAPP core with C1 are included partially in the figure. Shown are C1 p.(Val121Alafs*3) variant in dark red, C1 p.(His22_Lys24del) variant in light red, C1 subunit in yellow, C3 subunit in cyan, C4 subunit in light pink, and C6 subunit in hot pink.

**Figure 5 cells-13-01457-f005:**
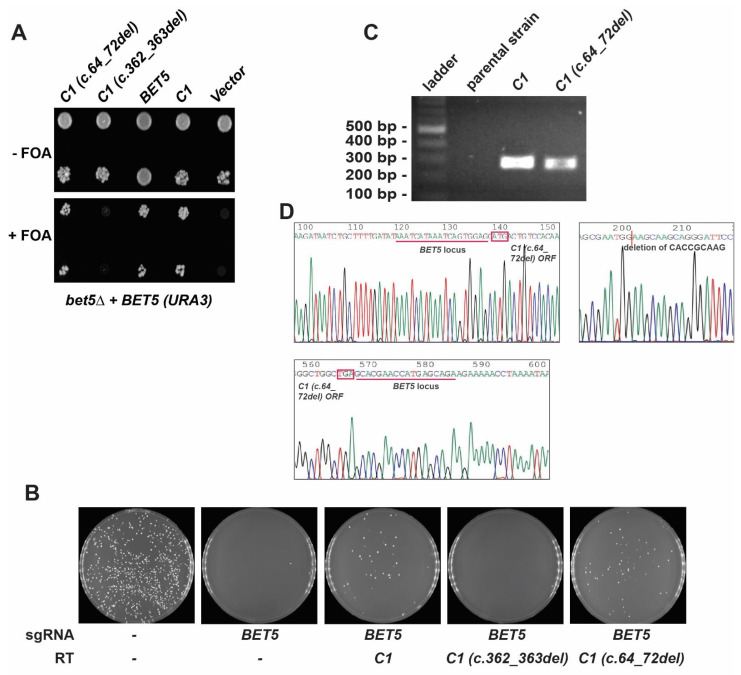
*C1 (c.64_72del)* but not *C1 (c.362_363del)* can compensate for the loss of *BET5* in yeast. (**A**) Haploid yeast strain *bet5*∆ kept alive with *URA3*-based *BET5* was transformed with an empty plasmid, with a vector containing *BET5* under its endogenous promoter or with vectors expressing either *C1*, *C1 (c.362_363del)*, or *C1 (c.64_72del)* under the control of the *ADH1* promoter. Transformed cells were plated on 5-FOA to counter-select for the *URA3*-based plasmids. (**B**) Yeast cells were transformed with either *C1*, *C1 (c.362_363del)*, or *C1 (c.64_72del)* repair templates in the presence of the self-containing CRISPR/Cas9 plasmid. In one control (left-most), the cells were not transformed with guide RNA or repair template to evaluate the transformation efficiency. A second control (second from left) used only gRNA to demonstrate the lethality of a DSB. The presence or absence of growth in the remaining plates is an indication of rescue or no rescue from the DSB lethality of the Cas9 nuclease when an appropriate repair template is provided. (**C**) An image from the colony PCR products of the transformation of yeast strains with *C1 (c.64_72del)* using humanized *C1* strain and parental strain as controls. The expected band size for each successful transformation is 274 bp for *C1* and 265 bp for *C1 (c.64_72del)*. (**D**) Electropherogram of the humanized yeast locus with *C1 (c.64_72del)*. The start and stop codons are shown in red boxes, the location of the 9-nucleotide deletion for *C1 (c.64_72del)* ORF is indicated and the corresponding UTR regions for the *BET5* yeast locus are underlined in red.

**Figure 6 cells-13-01457-f006:**
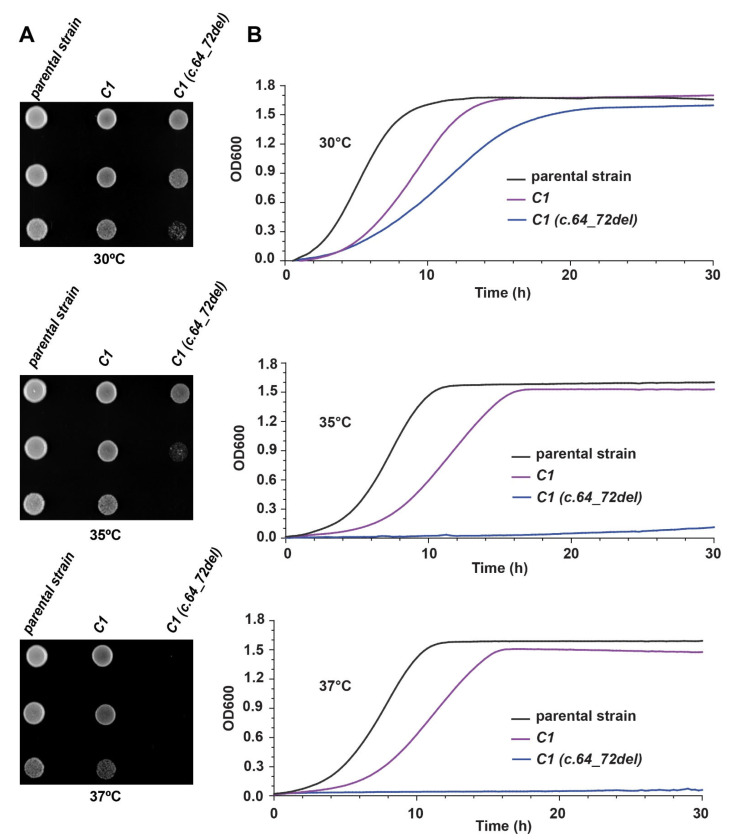
Growth efficiency of humanized *C1 (c.64_72del)* strain in solid and liquid media. (**A**) Images from the plates of parental strain (MSY135) and humanized yeast strains with *C1* and *C1 (c.64_72del)*. Serial dilutions of the liquid cultures were spotted on YPD plates and incubated for 48 h at three different temperatures, 30 °C, 35 °C, and 37 °C. (**B**) Representation of the growth curves in liquid medium for MSY135 and humanized yeast strains with *C1* and *C1 (c.64_72del)* at 30 °C, 35 °C, and 37 °C. Shown are the averages of OD600 for three biological replicates.

**Figure 7 cells-13-01457-f007:**
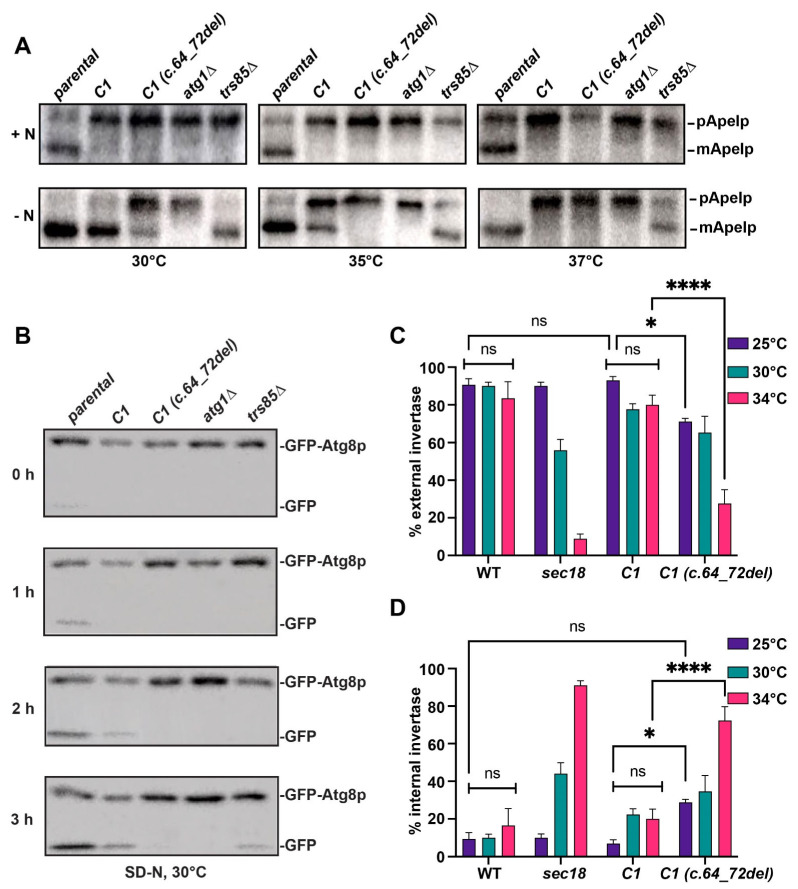
Non-selective autophagy and secretion are affected in the humanized *C1 (c.64_72del)* strain. (**A**) The parental strain MSY135 and the humanized strains with *C1* and *C1 (c.64_72del)* were grown in rich (+N panel) or nitrogen-starved (-N panel) media to induce autophagy at three different temperatures, 30 °C, 35 °C, and 37 °C, as described in the Section 2. The mutant strains *atg1*∆ and *trs85*∆ were used as controls to show defects in autophagy. Equal amounts of protein were fractionated by SDS–PAGE and analyzed by Western blot analysis using anti-Ape1p IgG. (**B**) The parental strain MSY135 and the humanized strains with *C1* and *C1 (c.64_72del)* expressing plasmid-based GFP-Atg8 under the control of the endogenous *ATG8* promoter were grown in nitrogen-starved medium (SD-N) to induce autophagy at 30 °C. The mutant strains *atg1*∆ and *trs85*∆ were used as controls to show defects in autophagy. At the indicated times, aliquots were removed and protein extracts were prepared and fractionated by SDS-PAGE, and analyzed by Western blot analysis using anti-GFP antibody. (**C**,**D**) Yeast cells (wild-type, *sec18*, humanized *C1* and *C1 (c.64_72del)*) were grown in medium with 0.05% glucose and 2% sucrose to induce expression of the *SUC2* gene that produces the secreted invertase enzyme. Induction was allowed to proceed for 1 h, at which time cells were collected and analyzed before or after lysis to determine percent external (**C**) and internal (**D**) invertase. Percentage external was calculated as (whole cell/lysed cell) × 100, and percentage internal was calculated as (lysed cell–whole cell/lysed cell) × 100. N = 8 over three biological replicates and error bars indicate SEM. Significance was calculated using a one way ANOVA with a Bonferroni post hoc correction. * indicates *p* < 0.05; **** indicates *p* < 0.0001; ns = not significant.

**Figure 8 cells-13-01457-f008:**
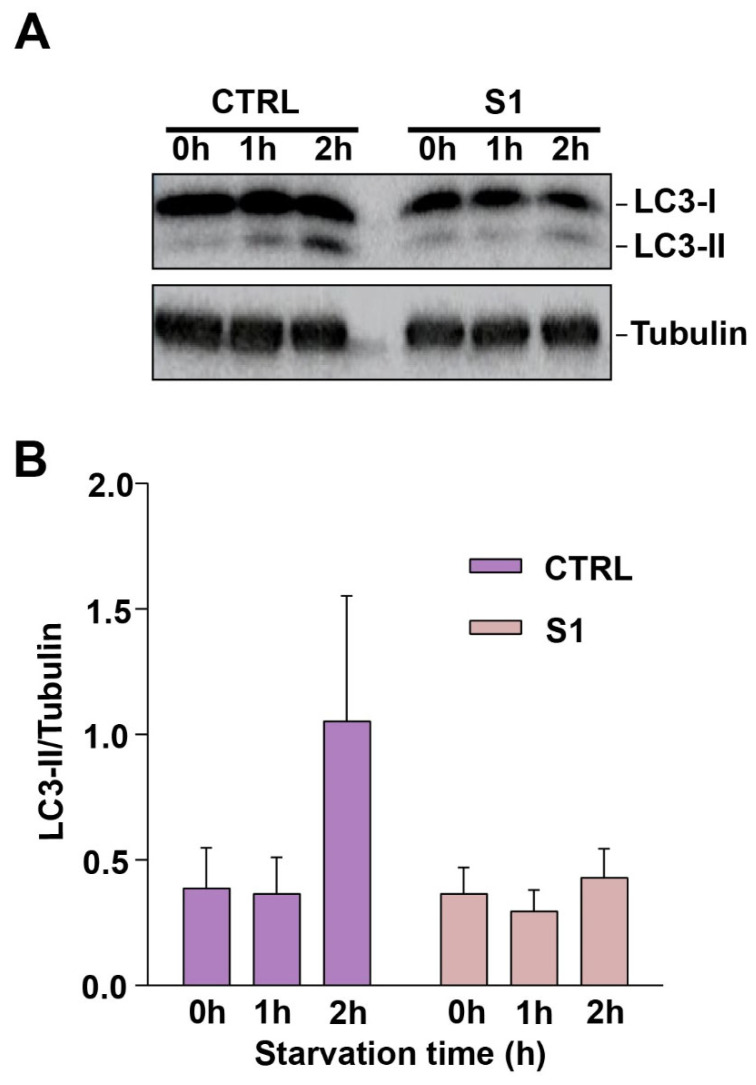
Patient fibroblasts harboring the compound heterozygous *TRAPPC1* mutation show a defect in autophagy. (**A**) Control (CTRL#1) and patient fibroblasts (S1) were starved (EBSS medium) for up to 2 h and cell lysates were collected at 0 h, 1 h, and 2 h time points. Shown is a representative Western blot where the expected bands for LC3-I, LC3-II, and tubulin (loading control) are indicated. LC3-I migrates at ~17 kDa and LC3-II migrates at ~15 kDa. (**B**) Quantification of the Western blots with LC3-II signal normalized to the tubulin loading control. Quantification was performed with ImageJ. The error bars indicate SEM.

**Figure 9 cells-13-01457-f009:**
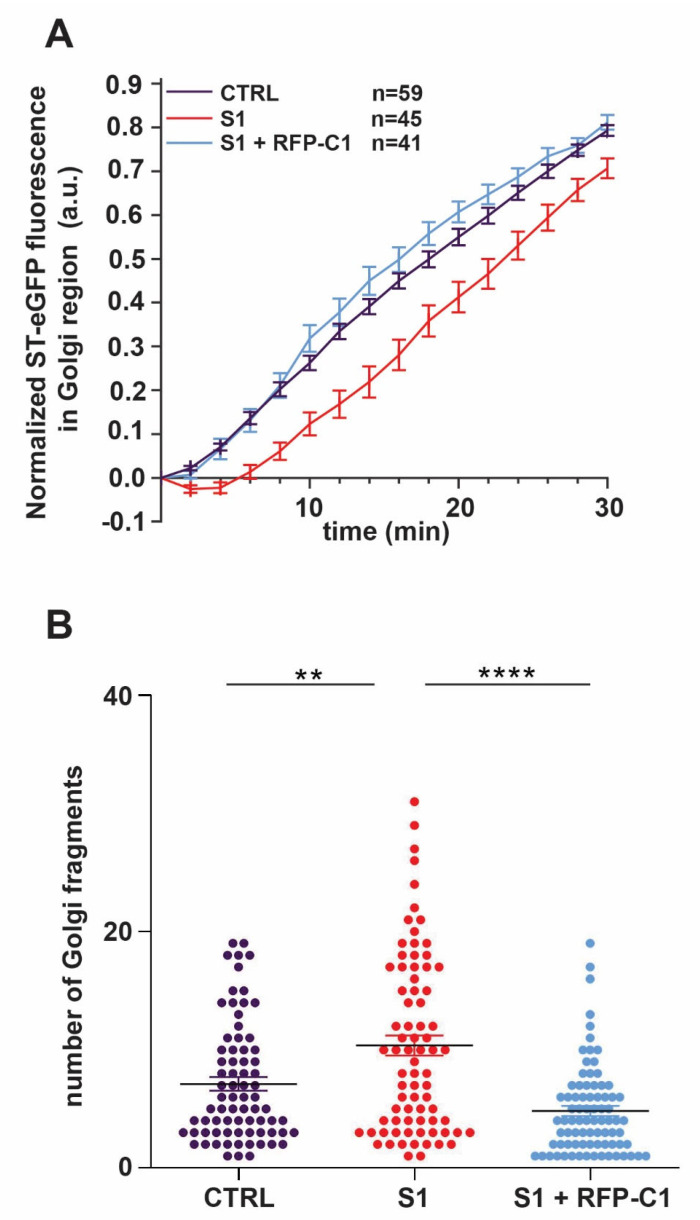
Trafficking from the ER-to-Golgi is delayed and Golgi morphology is altered in patient fibroblasts harboring a compound heterozygous *TRAPPC1* mutation. (**A**) The RUSH assay was performed on control, S1 patient fibroblasts, and S1 + RFP-C1 patient fibroblasts using ST-eGFP cargo protein. The cells were imaged every 2 min over a period of 30 min upon biotin addition, which causes the release of the cargo protein from the ER. Fluorescence intensity in the Golgi was quantified using ImageJ as described in Koehler et al., 2017. Representative images of the cells in each condition for the time points indicated are shown in Appendix A. (**B**) Control fibroblasts (CTRL), patient fibroblasts untransfected (S1), or fibroblasts transfected with a *C1* construct fused to RFP (S1 + RFP-C1) were fixed and stained for manosidase II to visualize Golgi and Hoechst to visualize the nucleus. The number of Golgi fragments per cell was quantified using Imaris as described in the Materials and Methods section. For estimating the statistical significance, one-way ANOVA was used with post hoc Tukey HSD analysis. The error bars indicate SEM. The N values for the control, S1, and S1 + RFP-C1 are 75, 78, and 81, respectively, in three different biological replicates. ** indicates *p* < 0.01; **** indicates *p* < 0.0001.

**Table 1 cells-13-01457-t001:** Mammalian and yeast TRAPP subunits. Mammalian subunits together with their yeast counterparts are listed and their presence in TRAPP II and/or TRAPP III is shown. Essential yeast TRAPP subunits are indicated as is the percentage of identity and similarity between yeast and human orthologs for the core subunits. Orthologs between yeast and mammals are listed on the same row. * TRAPPC13 is part of TRAPP III in mammals and Trs65p is part of TRAPP II in yeast.

Mammalian Subunits	Yeast Subunits	TRAPP Complex	Essential in Yeast?	%Identity/Similarity of Core Subunits
TRAPPC1	Bet5p	II/III	yes	33/54
TRAPPC2	Trs20p	II/III	yes	34/52
TRAPPC2L	Tca17p	II	no	24/43
TRAPPC3	Bet3p	II/III	yes	56/76
TRAPPC4	Trs23p	II/III	yes	29/44
TRAPPC5	Trs31p	II/III	yes	31/52
TRAPPC6A/C6B	Trs33p	II/III	no	35/51 and 33/47
TRAPPC8	Trs85p	III	no	-
TRAPPC9	Trs120p	II	yes	-
TRAPPC10	Trs130p	II	yes	-
TRAPPC11	-	III	-	-
TRAPPC12	-	III	-	-
TRAPPC13	Trs65	III/II *	no	-

**Table 2 cells-13-01457-t002:** Plasmids used in this study.

Plasmid	Construct or Backbone
MSB2	pRS315
MSB206	pRS313
MSB230	pRS313-TRS20
MSB1253	pRS416-GFP-ATG8
MSB1790	pmRFP-N1-TRAPPC1
MSB1791	pmRFP-C1-TRAPPC1
MSB1797	pRS413-ADH1-TRAPPC1
MSB1800	pRS413-ADH1-TRAPPC1 (c.362_363del)
MSB1801	pRS413-ADH1-TRAPPC1 (c.64_72del)
MSB1805	pCEN6-Cas9-GFP-KanMX
MSB1806	pCEN6-Cas9-GFP-KanMX BET5 sgRNA1
MSB1820	pCEN6-Cas9-GFP-KanMX TRS20 sgRNA1
MSB1821	pCEN6-Cas9-GFP-KanMX TRS23 sgRNA1
MSB1823	pCEN6-Cas9-GFP-KanMX BET3 sgRNA1
MSB1824	pCEN6-Cas9-GFP-KanMX TRS31 sgRNA1
MSB1826	pCEN6-Cas9-GFP-KanMX TCA17 sgRNA1
MSB1829	pCEN6-Cas9-GFP-KanMX TRS33 sgRNA1
MSB1839	pRS413-ADH1-TRAPPC5
MSB1840	pRS413-GPD-TRAPPC5
MSB1841	pRS423-GPD-TRAPPC5
MSB1855	pRS413-ADH1-TRAPPC3
MSB1856	pRS413-GPD-TRAPPC3
MSB1857	pRS423-GPD-TRAPPC3
MSB1858	pRS413-ADH1-TRAPPC4
MSB1859	pRS413-GPD-TRAPPC4
MSB1860	pRS423-GPD-TRAPPC4

**Table 3 cells-13-01457-t003:** Yeast strains used in this study.

Yeast Strain	Genotype	Source
MSY61	*MATα his3*Δ*1 leu2*Δ*0 ura3*Δ*0 MET15 bet3*∆*::KanMX pRS315-BET3*	Sacher Lab
MSY62	*MATα his3*Δ*1 leu2*Δ*0 ura3*Δ*0 MET15 trs23*∆*::KanMX pRS316-TRS23*	Sacher Lab
MSY64	*MATα his3*Δ*1 leu2*Δ*0 ura3*Δ*0 MET15 bet5*∆*::KanMX pRS316-BET5*	Sacher Lab
MSY65	*MATα his3*Δ*1 leu2*Δ*0 ura3*Δ*0 MET15 trs31*∆*::KanMX pRS316-TRS31*	Sacher Lab
MSY135	*MATα his3*Δ*1 leu2*Δ*0 lys2*Δ*0 ura3*Δ*0*	Sacher Lab
MSY362	*MATα his3*Δ*1 leu2*Δ*0 lys2*Δ*0 ura3*Δ*0 trs85*∆*::KanMX*	Titorenko Lab
MSY563	*MATα his3*Δ*1 leu2*Δ*0 ura3*Δ*MET15 trs23*Δ*::KanMX pRS315-TRS23 TRS85–3xHA::HIS3*	Sacher Lab
MSY706	*MATα his3*Δ*1 leu2*Δ*0 lys2*Δ*0 ura3*Δ*0 atg1*∆*::KanMX*	Brett Lab
MSY710	*MATα his3*Δ*1 leu2*Δ*0 lys2*Δ*0 ura3*Δ*0 atg1*∆*::KanMX pRS416-GFP-ATG8*	Sacher Lab
MSY740	*MATα his3*Δ*1 leu2*Δ*0 lys2*Δ*0 ura3*Δ*0 trs85*∆*::KanMX pRS416-GFP-ATG8*	Sacher Lab
MSY766	*MATα his3*Δ*1 leu2*Δ*0 lys2*Δ*0 ura3*Δ*0 pRS416-GFP-ATG8*	Sacher Lab
MSY901	*MATα his3*Δ*1 leu2*Δ*0 lys2*Δ*0 ura3*Δ*0 TRS130-3xHA::HIS3*	Sacher Lab
MSY950	*MATα his3*Δ*1 leu2*Δ*0 ura3*Δ*0 MET15 bet5*∆*::KanMX pRS316-BET5 pRS413-ADH1*	Zykaj E.
MSY951	*MATα his3*Δ*1 leu2*Δ*0 ura3*Δ*0 MET15 bet5*∆*::KanMX pRS316-BET5 pRS413-ADH1-TRAPPC1*	Zykaj E.
MSY952	*MATα his3*Δ*1 leu2*Δ*0 ura3*Δ*0 MET15 bet5*∆*::KanMX pRS316-BET5 pRS313-BET5*	Zykaj E.
MSY955	*MATα his3*Δ*1 leu2*Δ*0 lys2*Δ*0 ura3*Δ*0 bet5*∆ *with TRAPPC1*	Zykaj E.
MSY956	*MATα his3*Δ*1 leu2*Δ*0 lys2*Δ*0 ura3*Δ*0 bet5*∆ *with TRAPPC1* ∆*HRK (c.64_72del)*	Zykaj E.
MSY957	*MATα his3*Δ*1 leu2*Δ*0 lys2*Δ*0 ura3*Δ*0 trs20*∆ *with TRAPPC2*	Zykaj E.
MSY958	*MATα his3*Δ*1 leu2*Δ*0 lys2*Δ*0 ura3*Δ*0 TRS130-3xHA::HIS3 tca17*∆ *with TRAPPC2L*	Zykaj E.
MSY959	*MATα his3*Δ*1 leu2*Δ*0 lys2*Δ*0 ura3*Δ*0 TRS130-3XHA::HIS3 trs33*∆ *with TRAPPC6A*	Zykaj E.
MSY960	*MATα his3*Δ*1 leu2*Δ*0 lys2*Δ*0 ura3*Δ*0 TRS130-3XHA::HIS3 trs33*∆ *with TRAPPC6B*	Zykaj E.
MSY963	*MATα his3*Δ*1 leu2*Δ*0 ura3*Δ*0 MET15 trs31*∆*::KanMX pRS316-TRS31 pRS413-ADH1-TRAPPC5*	Zykaj E.
MSY964	*MATα his3*Δ*1 leu2*Δ*0 ura3*Δ*0 MET15 trs31*∆*::KanMX pRS316-TRS31 pRS413-GPD-TRAPPC5*	Zykaj E.
MSY965	*MATα his3*Δ*1 leu2*Δ*0 ura3*Δ*0 MET15 trs31*∆*::KanMX pRS316-TRS31 pRS423-GPD-TRAPPC5*	Zykaj E.
MSY967	*MATα his3*Δ*1 leu2*Δ*0 ura3*Δ*0 MET15 trs31*∆*::KanMX pRS316-TRS31 pRS315*	Zykaj E.
MSY968	*MATα his3*Δ*1 leu2*Δ*0 ura3*Δ*0 MET15 trs31*∆*::KanMX pRS316-TRS31 pRS315-TRS31*	Zykaj E.
MSY971	*MATα his3*Δ*1 leu2*Δ*0 ura3*Δ*0 MET15 trs23*∆*::KanMX pRS316-TRS23 pRS313*	Zykaj E.
MSY972	*MATα his3*Δ*1 leu2*Δ*0 ura3*Δ*0 MET15 trs23*∆*::KanMX pRS316-TRS23 pRS313-TRS23*	Zykaj E.
MSY973	*MATα his3*Δ*1 leu2*Δ*0 ura3*Δ*0 MET15 trs23*∆*::KanMX pRS316-TRS23 pRS413-ADH1-TRAPPC4*	Zykaj E.
MSY974	*MATα his3*Δ*1 leu2*Δ*0 ura3*Δ*0 MET15 trs23*∆*::KanMX pRS316-TRS23 pRS413-GPD-TRAPPC4*	Zykaj E.
MSY975	*MATα his3*Δ*1 leu2*Δ*0 ura3*Δ*0 MET15 trs23*∆*::KanMX pRS316-TRS23 pRS423-GPD-TRAPPC4*	Zykaj E.
MSY976	*MATα his3*Δ*1 leu2*Δ*0 ura3*Δ*0 MET15 bet3*∆*::KanMX pRS315-BET3 pRS313*	Zykaj E.
MSY977	*MATα his3*Δ*1 leu2*Δ*0 ura3*Δ*0 MET15 bet3*∆*::KanMX pRS315-BET3 pRS313–BET3*	Zykaj E.
MSY978	*MATα his3*Δ*1 leu2*Δ*0 ura3*Δ*0 MET15 bet3*∆*::KanMX pRS315-BET3 pRS413-ADH1-TRAPPC3*	Zykaj E.
MSY979	*MATα his3*Δ*1 leu2*Δ*0 ura3*Δ*0 MET15 bet3*∆*::KanMX pRS315-BET3 pRS413-GPD-TRAPPC3*	Zykaj E.
MSY980	*MATα his3*Δ*1 leu2*Δ*0 ura3*Δ*0 MET15 bet3*∆*::KanMX pRS315-BET3 pRS423-GPD-TRAPPC3*	Zykaj E.
MSY981	*MATα his3*Δ*1 leu2*Δ*0 lys2*Δ*0 ura3*Δ*0 bet5*∆ *with TRAPPC1* pRS416-GFP-ATG8	Zykaj E.
MSY982	*MATα his3*Δ*1 leu2*Δ*0 lys2*Δ*0 ura3*Δ*0 bet5*∆ *with TRAPPC1* ∆*HRK (c.64_72del) pRS416-GFP-ATG8*	Zykaj E.
MSY983	*MATα his3*Δ*1 leu2*Δ*0 lys2*Δ*0 ura3*Δ*0 TRS130-3xHA::HIS3 bet5*∆ *with TRAPPC1*	Zykaj E.
MSY984	*MATα his3*Δ*1 leu2*Δ*0 lys2*Δ*0 ura3*Δ*0 TRS130-3xHA::HIS3 bet5*∆ *with TRAPPC1* ∆*HRK (c.64_72del)*	Zykaj E.
MSY987	*MATα his3*Δ*1 leu2*Δ*0 lys2*Δ*0 ura3*Δ*0 tca17*∆ *with TRAPPC2L*	Zykaj E.
MSY988	*MATα his3*Δ*1 leu2*Δ*0 lys2*Δ*0 ura3*Δ*0 trs33*∆ *with TRAPPC6B*	Zykaj E.
MSY993	*MATα his3*Δ*1 leu2*Δ*0 lys2*Δ*0 ura3*Δ*0 trs33*∆ *with TRAPPC6A*	Zykaj E.

**Table 4 cells-13-01457-t004:** Oligonucleotides designed for each sgRNA targeting the yeast TRAPP genes.

Primer Name	Sequence
BET5 sgRNA1-Fp	GACTTTTTACACAAAAGCTCTCCAAA
BET5 sgRNA1-Rp	AAACTTTGGAGAGCTTTTGTGTAAAA
TRS20 sgRNA1-Fp	GACTTTACCAATGCAGAAAATCCACA
TRS20 sgRNA1-Rp	AAACTGTGGATTTTCTGCATTGGTAA
TCA17 sgRNA1-Fp	GACTTTGATTTACGTCCCTAATGAGG
TCA17 sgRNA1-Rp	AAACCCTCATTAGGGACGTAAATCAA
BET3 sgRNA1-Fp	GACTTTCTTCAAAGACCTCGATTGCG
BET3 sgRNA1-Rp	AAACCGCAATCGAGGTCTTTGAAGAA
TRS23 sgRNA1-Fp	GACTTTCTTGTAATAAACAAATCAGG
TRS23 sgRNA1-Rp	AAACCCTGATTTGTTTATTACAAGAA
TRS31 sgRNA1-Fp	GACTTTGCATCTGACCAACAATTCCC
TRS31 sgRNA1-Rp	AAACGGGAATTGTTGGTCAGATGCAA
TRS33 sgRNA1-Fp	GACTTTGGACCTTGTGGAGAAGATTG
TRS33 sgRNA1-Rp	AAACCAATCTTCTCCACAAGGTCCAA

**Table 5 cells-13-01457-t005:** Oligonucleotides designed for each repair template.

Primer Name	Sequence
TRAPPC1-Fp	AATACAAAAAAAAAATACAGAACTATCGTAAGATAATCTGCTTTTGATATAAATCATAAATCAGTGGAGCATGACTGTCCACAACCTGTAC
TRAPPC1-Rp	TATACGATGAATGCAATTCAATTCATCGCTTCTATTTATTTTAGGTTTTTCTTCTGCTCATGGTTCGTGCTCAGCCAGCCCGGGCGGAGAAG
TRAPPC2-Fp	AAGAAAAAAAAGAGAAGGTAAACACTAATACGAATAGAGAATACAGAAAAAACATACAAAGCACATTGAGATGTCTGGGAGCTTCTACTTTG
TRAPPC2-Rp	AAAAAAAAAAAAATACATACTACATATACATATACGCCATAAAAAATCTCTGCATCTATCTTATTTCCCATCAGCTTAAAAGGTGTTTCTTC
TRAPPC2L-Fp	AAGTTATTGAGTTGAAAATTGAGTAAATATTCTCTCCAAAAATCAAATTGTACGTTTATATCACCAAACTATGGCGGTGTGCATCGCGGTG
TRAPPC2L-Rp	GCTCTCACTCCAAAGCAGTGACATTAAATTTTGCTTTCATTCGTGAAAGAATCGACTAATAAATTATCATTCAGCACACCTGTATCATCATC
TRAPPC3-Fp	CCATAGCTATAGAGATGGGAATGCAGAGTAAAGCTTCAAGTGTTCATTGATAACCAAAACTGGGTCAAAAATGTCGAGGCAGGCGAACCGTG
TRAPPC3-Rp	AAATAATGTATACCTAAATAACGACACCAATAATAACAAAAATCACGAAAAAAAAAAGCTTACATGTCCATTATTCCTCTCCAGCTGGAAG
TRAPPC4-Fp	AAAAGGAATCTGCCTTTGCATAAGTTCAAAAGTGCAATTTTAGTGTTGGATTTAAACGGGAAAAATTGAAATGGCGATTTTTAGTGTGTATG
TRAPPC4-Rp	CTTAGTTCTAGAAGTACGTAGTATTTATTTTCTTGGTGTGAATGCGTTTATCTTCCGATGGCGCGTGCGTCTATGACCCAGGTCCAAAAG
TRAPPC5-Fp	AGGACAAGAGACACGGAAGCGACAGAAAGACAGGGAGTACTATCACCATCCATAATACTGGAGTAACATTATGGAGGCGCGCTTCACGCGC
TRAPPC5-Rp	TCAACAGTAAATATACAAACTCTTTATCCTTTTCTTGGTTTTAGGTATTTACCGATGTTTTGAGATGATATCAGCGGCCCTCCAGGGCCCG
TRAPPC6A-Fp	AGCTGAGAGTGAAGACCTCATGCCATAAAAGAAAACATACACCGGTTATAAAGTTGGAAGATAAACAATTATGGCGGATACTGTGTTGTTTG
TRAPPC6A-Rp	TCTTTTATATACACACTTATATCTATCTATATGTCGATGTACATTCTTAGAACAAAAATCTGTCGGACCTTTAGGATTTCGGAATCACCACC
TRAPPC6B-Fp	AGCTGAGAGTGAAGACCTCATGCCATAAAAGAAAACATACACCGGTTATAAAGTTGGAAGATAAACAATTATGGCGGATGAGGCGTTGTTTTTG
TRAPPC6B-Rp	TCTTTTATATACACACTTATATCTATCTATATGTCGATGTACATTCTTAGAACAAAAATCTGTCGGACCTCTACAGCTTCTGTATCATCAC

**Table 6 cells-13-01457-t006:** Primers used to confirm humanization of yeast.

Primer Name	Sequence
TRAPPC1-Fp	GTCCTTTACTATTGACTAGTG
TRAPPC1-Rp	CATCCCGTACATCAGCTTATAC
TRAPPC2-Fp	CCCTTTACCTCTTCAAAGCCT
TRAPPC2-Rp	AAGTACATGTTGTTCGATAGC
TRAPPC2L-Fp	AAGTTCTGCCGGAAGAGCTCATC
TRAPPC2L-Rp	ATTGCGGAGATCTTCTCATCC
TRAPPC6A-Fp	GGAATCTGGTATGCACTCAC
TRAPPC6A-Rp	TGTTGTCTTGCAGGACGTAG
TRAPPC6B-Fp	GGAATCTGGTATGCACTCAC
TRAPPC6B-Rp	AACTCATCCTTGAACCTTGCAG

## Data Availability

Data are available from the authors upon request.

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
