# Peer review of "A Humanized Yeast Model for Studying TRAPP Complex Mutations; Proof-of-Concept Using Variants from an Individual with a TRAPPC1-Associated Neurodevelopmental Syndrome"

_cells, 2024, doi:10.3390/cells13171457_

Round 1

Reviewer 1 Report

Comments and Suggestions for Authors

This manuscript describes a humanized yeast model for investigating TRAPPC1-associated neurodevelopmental syndrome. The researchers use CRISPR/Cas9 technology to knock out yeast endogenous TRAPP complex components and insert human wild-type or variants from patients with TRAPP complex genes. Finally, they explore the variant's function in cell growth, autophagy, and ER-to-Golgi trafficking. They indeed construct a humanized model to study the gene variant and provide some functional investigation to prove the defect of this variant. I appreciate their work and only have some minor suggestions.

(1) The introduction in describing the rare disease model study is too long. I suggest shortening the description from lines 107-169.

(2) Because the analyses of autophagy and ER-to-Golgi trafficking were used in the results, I suggest adding more description about the functional effects of TRAPP on autophagy and ER-to-Golgi trafficking (extending line 84, introduce Ape1p, Atg8p, Atg1 these proteins).

(3) I suggest adding some discussion to interpret the association between neurodevelopmental syndrome, autophagy's defect, and ER-to-Golgi trafficking.

(4) Please correct the label of Figure 5A. 

Author Response

This manuscript describes a humanized yeast model for investigating TRAPPC1-associated neurodevelopmental syndrome. The researchers use CRISPR/Cas9 technology to knock out yeast endogenous TRAPP complex components and insert human wild-type or variants from patients with TRAPP complex genes. Finally, they explore the variant's function in cell growth, autophagy, and ER-to-Golgi trafficking. They indeed construct a humanized model to study the gene variant and provide some functional investigation to prove the defect of this variant. I appreciate their work and only have some minor suggestions.

(1) The introduction in describing the rare disease model study is too long. I suggest shortening the description from lines 107-169.
We have shortened the Introduction as per this reviewer’s comment.

(2) Because the analyses of autophagy and ER-to-Golgi trafficking were used in the results, I suggest adding more description about the functional effects of TRAPP on autophagy and ER-to-Golgi trafficking (extending line 84, introduce Ape1p, Atg8p, Atg1 these proteins).
We have now elaborated on the role of TRAPP in ER-to-Golgi transport and added some information on the autophagy proteins studied.

(3) I suggest adding some discussion to interpret the association between neurodevelopmental syndrome, autophagy's defect, and ER-to-Golgi trafficking.
We have added some text in the discussion linking both autophagy and membrane traffic to neurodevelopmental diseases.

 (4) Please correct the label of Figure 5A.
Thank you for pointing this out. We have corrected the labeling in panel A of Figure 5.

Reviewer 2 Report

Comments and Suggestions for Authors

In this interesting manuscript, the authors develop a humanized yeast model of the TRAPP complexes. They do this by replacing yeast proteins Bet5p, Trs20p, Tca17p, and Trs33p with their human orthologs, and TRAPPC1, TRAPPC2, TRAPPC2L, and TRAPPC6A or TRAPPC6B

They then use this system to investigate the impact of two human disease variants in TRAPPC1.  They find that one of these variants is non-functional and the other is conditional-lethal (temperature-sensitive). They further investigate this conditional mutant and characterize its impacts on secretion and autophagy.

This system therefore provides a useful new model for investigating the effects of human variants that can be used as a diagnostic tool.

Overall I think this work is interesting and the data support the conclusions.  I have just one suggestion for improvement:  The authors have not determined whether the TRAPPC1 variants are actually expressed and stable in yeast (i.e., there is no assessment of protein levels by western blot).  It is possible the variants are not expressed or stable in yeast, and this may or may not correlate with their expression and stability in humans. The authors should address these possibilities in the text.

Author Response

In this interesting manuscript, the authors develop a humanized yeast model of the TRAPP complexes. They do this by replacing yeast proteins Bet5p, Trs20p, Tca17p, and Trs33p with their human orthologs, and TRAPPC1, TRAPPC2, TRAPPC2L, and TRAPPC6A or TRAPPC6B

 They then use this system to investigate the impact of two human disease variants in TRAPPC1.  They find that one of these variants is non-functional and the other is conditional-lethal (temperature-sensitive). They further investigate this conditional mutant and characterize its impacts on secretion and autophagy.

 This system therefore provides a useful new model for investigating the effects of human variants that can be used as a diagnostic tool.

 Overall I think this work is interesting and the data support the conclusions.  I have just one suggestion for improvement:  The authors have not determined whether the TRAPPC1 variants are actually expressed and stable in yeast (i.e., there is no assessment of protein levels by western blot).  It is possible the variants are not expressed or stable in yeast, and this may or may not correlate with their expression and stability in humans. The authors should address these possibilities in the text.
We have indeed attempted this but the antibody we purchased was simply not good for western blotting and did not provide conclusive results. We suspect that the protein is indeed expressed since the yeast homologue Bet5p is essential and the TRAPP core would not be expected to form in its absence. We have added this idea to the discussion section.